# Is Interstitial Chemotherapy with Carmustine (BCNU) Wafers Effective against Local Recurrence of Glioblastoma? A Pharmacokinetic Study by Measurement of BCNU in the Tumor Resection Cavity

**DOI:** 10.3390/brainsci12050567

**Published:** 2022-04-28

**Authors:** Takanori Ohnishi, Daisuke Yamashita, Akihiro Inoue, Satoshi Suehiro, Shiro Ohue, Takeharu Kunieda

**Affiliations:** 1Department of Neurosurgery, Washoukai Sadamoto Hospital, Matsuyama 790-0052, Ehime, Japan; 2Department of Neurosurgery, Ehime University Graduate School of Medicine, Toon 791-0295, Ehime, Japan; yamashita.daisuke.af@ehime-u.ac.jp (D.Y.); iakihiro3@gmail.com (A.I.); satoshis@m.ehime-u.ac.jp (S.S.); takeharukunieda@gmail.com (T.K.); 3Department of Neurosurgery, Ehime Prefectural Central Hospital, Matsuyama 790-0024, Ehime, Japan; c-sohue@eph.pref.ehime.jp

**Keywords:** glioblastoma, BCNU wafer, CSF concentration of BCNU, invasion, tumor recurrence

## Abstract

The effectiveness of carmustine (BCNU) wafers on local recurrence of glioblastoma (GBM) remains contentious. We investigated the accumulating high-dose effects of BCNU released from the wafers on the survival of GBM patients by measuring BCNU concentration in the resection cavity of GBM over time. BCNU wafers (Gliadel^®^) were implanted with an Ommaya device in 15 patients, including 12 patients with GBM. BCNU concentrations in the tumor resection cavity were measured for 30 days postoperatively. The area under the curve (AUC)_all_ was calculated from BCNU concentration curves, and the relationships between AUC_all_ and survival, tumor phenotypes on MRI, and recurrence patterns were analyzed. The BCNU concentration was maximal 1 h postoperatively, rapidly decreased within 24 h, and remained relatively high for 7 days. GBM patients were classified into two groups: early recurrence (ER) and late or no recurrence (LN), using median progression-free survival as the cut-off. AUC_all_ tended to be lower in the ER group than in the LN group, but the difference was not significant. MRI revealed that all patients in the ER group had highly invasive GBMs, whereas all patients in the LN group had less-invasive GBMs. A total of 9 patients experienced recurrence, with 6 local, 2 diffuse, and 1 disseminated patterns. No differences in AUC_all_ were seen between local and non-local recurrence groups. Total BCNU concentrations did not correlate with tumor progression or survival. However, a high concentration of BCNU may have potential to provide some survival benefit for less-invasive type GBM.

## 1. Introduction

Malignant glioma is highly resistant to radiochemotherapy, resulting in early tumor recurrence and short survival. The goal of treatment for malignant glioma is maximum resection of the tumor while maintaining normal neurological functions, followed by radiation and chemotherapy with mainly temozolomide (TMZ) [1]. However, despite aggressive surgical resection using multimodal navigation systems, malignant glioma, especially glioblastoma (GBM), remains incurable with poor prognosis because of the highly infiltrative nature of GBM [2].

Since 2013, bis-chloroethyl nitrosourea (BCNU) (carmustine)-impregnated wafers (Gliadel^®^) have been used as chemotherapeutic agents and as interstitial adjuvant therapy in many countries, including Japan [3,4]. Because local recurrence is thought to be the most frequent pattern of treatment failure with the Stupp regimen, these wafers were expected to be useful for local control of GBM [2]. Many studies about the effectiveness of BCNU wafers have been reported [5,6] but the favorable effects of BCNU wafers on prognosis are still disputed.

Here, to investigate whether BCNU wafers are effective for controlling tumor progression, whether anti-tumor effects of BCNU wafers depend on the accumulating concentration of BCNU, and what type of tumors show a resistance to BCNU wafers, we measured the BCNU concentration in the resection cavity up to 30 days after implantation of wafers by assessing cerebral spinal fluid (CSF) samples and analyzed the relationship between the total concentrations of BCNU and survival times in patients with GBM. We also investigated whether a high concentration of BCNU that was locally released into the area adjacent to the tumor resection cavity affected the pattern of tumor recurrence. In addition, to investigate a phenotype of GBM showing resistance to high-dose BCNU, we examined imaging features of GBM on magnetic resonance imaging (MRI) and ^11^C-methionine (Met)-positron emission tomography (PET) and analyzed the correlation between imaging features and anti-tumor effect of high-dose BCNU. We have previously reported that imaging features of GBM on MRI and Met-PET well correlated with the expression level of CD44 in the tumor tissues of GBM [7,8,9]. In the present study, we examined gene expression of CD44 in the tumor tissues of GBM to identify the features of specific phenotypes of GBM. These studies provide crucial information for obtaining optimal efficacy in the treatment with BCNU wafers in GBM.

## 2. Materials and Methods

All procedures performed in studies involving human participants were done in accordance with the ethical standards of the institutional and/or national research committee and the 1964 Helsinki Declaration and its later amendments or comparable ethical standards. The present study was approved by the Ethics Committee for Clinical Research of Ehime University Hospital (no. 1410014).

### 2.1. Patients and Study Design

A total of 18 patients whose histologies were verified as malignant gliomas (14 GBMs, 4 astrocytoma IDH mutants, grade 3 (AMG3)) who were treated according to the same treatment protocol except for implantation of BCNU wafers at the Department of Neurosurgery of Ehime University Hospital between November 2014 and December 2016 were enrolled. Among them, 3 patients (2 GBM and 1 AMG3) were assigned to the control group in this study. Informed consent was obtained from all individual participants enrolled in the study after the risk of the surgical procedure and the potential risks of microsurgery and chemoradiotherapy, including implantation of BCNU wafers (Gliadel^®^) were explained. All patients underwent craniotomy for tumor resection. After the tumor was resected, BCNU wafers were implanted on the surface of the tumor bed. Eight BCNU wafers were placed to closely cover the surface of the tumor resection cavity in each case and secured with Surgicel absorbable hemostat (Surgicel) (Johnson & Johnson, Tokyo, Japan) to prevent the wafers from floating up. If the ventricles were opened, they were closed as much as possible with Gelfoam^®^ (Pfizer Japan, Inc., Tokyo, Japan) and fibrin glue. Then, an Ommaya reservoir was placed under the scalp for obtaining CSF samples in the cavity. After tumor resection, radiotherapy (60 Gy) and chemotherapy with TMZ in accordance with the Stupp protocol were performed [10]. At 1 h, 1 day, 3 days, 7 days, 14 days, and 30 days postoperatively, CSF samples in the cavity were taken. Unchanged BCNU concentrations in the CSF and the number of cells and total protein concentrations in the CSF in the cavity were measured as detailed below. From the time course curves of BCNU concentrations from day 0 to day 30, the area under the curve (AUC)_all_ was calculated in each patient. Then, correlations between the AUC_all_ of BCNU in each patient and survival times, recurrence, and phenotypes on MRI in the patients with GBM were analyzed.

### 2.2. Measurement of the Concentration of BCNU in the CSF within the Tumor Resection Cavity

The concentrations of BCNU were obtained by quantitative measurement of intact BCNU in the CSF within the tumor cavity with liquid chromatography tandem mass spectrometry (LC–MS/MS), which was performed by ADME & Tox. Research Institute in Sekisui Medical Co., Ltd. (Ibaragi, Japan). Both validation of measurement methods for intact BCNU and measurement of intact BCNU in the CSF were performed using an LC–MS/MS system (MS/MS: API4000 (Sciex, Tokyo, Japan), LC: LC-10AD (Shimazu Corp., Kyoto, Japan)). The lower limit of detection of BCNU was 0.1 ng/mL.

### 2.3. Measurement of Protein Concentration and Number of Cells in the CSF within the Tumor Cavity

The concentration of proteins in the CSF in the tumor cavity was measured with the pyrogallol red method [8]. The number of cells within the tumor cavity was counted using a TBA-FX8 automated clinical chemistry analyzer (Canon Medical Systems, Tochigi, Japan), and classification of the cell types was performed following Samson staining.

### 2.4. Imaging Studies

MRI was performed using a 3T scanner (Achieva, Philips, Best, The Netherlands) with a standard head coil. Axial, coronal, and sagittal T1-weighted images were obtained with a slice thickness of 2 mm before and after intravenous administration of gadolinium-diethylenetriamine pentaacetic acid (0.1 mmol/kg). Axial fluid attenuated inversion recovery images were also obtained. PET studies were performed in a three-dimensional acquisition mode. ^11^C-Met-PET data were acquired for 20 min, beginning 20 min after administration of 5 MBq/kg body weight Met.

### 2.5. RNA Isolation and Quantitative Real-Time RT-PCR (qRT-PCR)

To investigate expression of CD44 in the tumor tissues of GBM, quantitative real-time RT-PCR was performed. Total RNA was extracted from the tissue of each tumor sample (both core and periphery) using ISOGEN (Nippon Gene, Tokyo, Japan) according to the manufacturer’s instructions. Complementary deoxyribonucleic acid (cDNA) was synthesized using ReverTra Ace qPCR RT Master Mix with a gDNA remover kit (Toyobo). qPCR analysis was performed using Fast Start Universal SYBR Green Master Mix (Roche Diagnostic Japan) with an MJ Mini instrument (BioRad, Hercules, CA, USA). All gene-specific mRNA expression values were normalized relative to the expression level of *GAPDH*, a housekeeping (reference) gene encoding glyceraldehyde-3-phosphate dehydrogenase. Quantification of gene expression was performed using ΔCt values, wherein ΔCt is defined as the difference between the target and reference gene Ct values. All primer sequences are listed in Appendix A.

### 2.6. Statistical Analysis

Values are expressed as the mean ± standard deviation, and the data were compared using the Student’s *t*-test (unpaired). Comparisons of data for more than two groups were carried out using one-way analysis of variance with the Tukey post hoc test. Kaplan–Meier plots were generated to estimate unadjusted time-to-event variables. The log-rank test was performed to assess the statistical significance of differences between groups. Spearman correlation analysis was performed to examine correlations for nonparametric data. Significance was set at *p* < 0.05. All analyses were performed using Office Excel 2016 software (Microsoft^®^, Redmond, WA, USA).

## 3. Results

### 3.1. Patient Characteristics

The patient characteristics are summarized in Table 1. Eighteen patients were enrolled in this study. All patients were newly diagnosed and primary therapeutic cases. Of 18 patients, 15 patients underwent craniotomy for tumor resection and implantation of BCNU wafers in the resected space with placing an Ommaya device, followed by radiotherapy (60 Gy) and chemotherapy with TMZ in accordance with the Stupp protocol [10]. The remaining three patients, who were treated under the same protocol including only placement of the Ommaya device without implantation of BCNU wafers, were used as controls. The resected tumors were verified by evaluating the molecular and histopathological analysis according to the World Health Organization classification 2021 [11]. The presence of hotspot mutations in isocitrate dehydrogenase I (IDH-1) (R132H) was evaluated by Sanger sequencing. The status of methylation of the O(6)-methylguanine-DNA methyltransferase (MGMT) promoter was analyzed by quantitative methylation-specific PCR after bisulfate modification of genomic DNA (using DNA Methylation Detection kit, Cosmmo Bio) and immunohistochemistry using a mouse monoclonal anti-MGMT antibody (MT3.1; Millipore). Immunohistochemistry was performed as described previously [7,12]. Evaluation of the PCR analysis was performed by using a cutoff value not less than 1% for MGMT promoter methylation [13]. The results of these analyses disclosed that 14 were GBMs, IDH-wild, grade 4, and 4 were astrocytomas, IDH-mutant, grade 3 (AMG3s). Moreover, 6 of the 14 patients with GBM and 2 of the 4 patients with AMG3 had MGMT promotor methylation (Table 1). No GBMs had IDH-1 mutation, while all AMG3s had mutation in IDH-1. The extent of resection was evaluated by volumetric analysis on MRI before and after surgery, as previously described [14]. Gross total resection (100% resection of the tumor volume), subtotal resection (95 to 100% resection), and partial resection (<95% resection) were achieved in 12 (66.7%), 3 (16.7%), and 3 patients (16.7%), respectively. No critical adverse effects were observed following implantation of either BCNU wafers or the Ommaya device, including infection and CSF leakage. A ventricular opening was accidentally created during the course of tumor resection in 1 patient (no. 9). Although this opening in the ventricular wall was tightly sealed, the case was excluded from the present analysis.

### 3.2. BCNU Concentration after Implantation of Wafers

Time course curves of BCNU concentrations in the tumor resection cavity of 14 patients with malignant glioma who were treated with BCNU wafers are shown in Figure 1a. The data of the measurement of BCNU concentrations of up to 30 days in these patients are summarized in Appendix A. At 30 days postoperatively, all patients showed undetectable or very low levels of BCNU concentrations (mean: 0.28 ± 0.12 ng/mL). Thus, we present the concentration curves based on the data from postoperative day 0 to day 14. In addition, as the values of BCNU concentrations varied widely and showed a lognormal-like distribution, the median BCNU concentrations were evaluated in addition to calculating the mean BCNU concentrations at each sampling time (Figure 1b). All mean BCNU concentrations were higher than the median BCNU concentrations at each sampling time but showed almost the same time course as the median BCNU concentrations. The BCNU concentration was the highest (mean/median: 697/560 ng/mL) at day 0 (1 h), rapidly decreased to 220/196 ng/mL within 24 h, was maintained at more than 176/124 ng/mL for 7 days, and then decreased to about 14.9/5.8 ng/mL at 14 days after implantation (Figure 1b). We found no significant difference in the BCNU concentration related to tumor volume, volume of the resection cavity, or existence of residual tumor.

### 3.3. Calculation of the AUC_all_ and the Relationship between AUC_all_ and Tumor Progression

To evaluate the total concentrations of BCNU that accumulated in the CSF in the cavity for 30 days, AUC_all_ was calculated using the noncompartmental analysis program of Phoenix WinNonlin 6.1 (Certara, L.P.; St. Louis, MO, USA) in each patient (Table 1). We found no correlation between overall survival (OS) time and AUC_all_ in 11 GBM patients (Figure 2a). Thus, we investigated whether a difference was present in the total concentrations of BCNU (AUC_all_ of BCNU) between patients with early progression and patients with late or no progression. The progression survival curve of 11 GBM patients treated with BCNU wafers is presented in Figure 2b, and the median progression-free survival (PFS) time was 9 months. Using the median PFS on BCNU wafer therapy as a cut-off value between early recurrence and late or no recurrence, the patients with PFS times < 9 months were classified in the early recurrence (ER) group, whereas those with PFS times ≥ 9 months were included in the late or no recurrence (LN) group. Each group included the following patients. ER: G1, G3, G8, G11, and G13 (five patients) and LN: G2, G4, G5, G6, G7, and G12 (six patients) (Table 1). The LN group showed a significantly longer OS time (48.7 months) and PFS time (35 months) than the ER group (OS time: 6.8 months, PFS time: 4.8 months) (OS, PFS; *p* = 0.0007) (Figure 2c,d). As to prognostic factors for these two groups, age, sex, KPS score and EOR were not significantly different between ER and LN groups. In addition, all GBM patients did not show IDH-I mutation. Methylation of MGMT promotor was positive in three in the ER group and two in the LN group. Statistical analysis of MGMT promotor methylation revealed no significant difference between ER and LN groups.

### 3.4. Relationship between Total BCNU Concentration (AUC_all_) and Tumor Control

The mean AUC_all_ of BCNU concentrations in the LN group (2456.7 ± 1405.7 ng·day/mL) was greater than that in the ER group (1849 ± 522.4 ng·day/mL), but the difference between these two groups was not significant (*p* = 0.393) (Figure 3).

### 3.5. Imaging Characteristics of the 12 Patients with GBM

We previously classified GBM into two types according to MRI combined with PET imaging [7,9]. These included a highly invasive type, in which tumors are heterogeneously enhanced, show an irregular tumor margin, and present diffuse peritumoral edema (highly invasive type GBM), and a low-invasive and relatively well-demarcated type, in which tumors show intense and relatively homogeneous enhancement with focal edema (low invasive/demarcated type GBM). Details of the criteria of these phenotypes in GBM on MRI and PET were previously described [9]. Analysis of phenotypic features on MRI in the 12 GBMs revealed that all patients in the ER group showed the highly invasive type on MRI, whereas all patients in the LN group presented the low-invasive and demarcated type on MRI (Table 1, Appendix A).

### 3.6. Expression of mRNA of CD44 in 12 Patients with GBM

We have demonstrated that glioma stem-like cells expressing CD44 to a high level at the tumor invasion zone in the periphery of GBM. In addition, highly invasive type of GBM presents much higher expression of CD44 than low-invasive type GBM [7]. Consequently, we examined expression of CD44 in the patients of two groups. We measured the mRNA expression of the gene encoding the stem cell marker CD44 in the tumor core and periphery, and calculated the ratio of amounts of CD44 expression in the tumor periphery to those in the tumor core (P/C ratio). We confirmed that the mean value of the P/C ratio of CD44 expression in the GBM of the high-invasive type (all patients in the ER group) was significantly higher than that in the GBM of the low-invasive type (all patients in the LN group) (HI/ER: 11.12 ± 4.18 (mean ± SD), LI/LN: 1.12 ± 0.42, *p* = 0.00017) (Figure 4).

### 3.7. Relationship between the BCNU Concentration and Tumor Recurrence

Nine of eleven GBM patients presented with tumor recurrence. Tumor growth patterns at the time of tumor recurrence were classified into three types based on the MRI features by considering the positional relationship between the recurrent tumor and the resection cavity. These included tumors recurring from the edge of the tumor resection cavity (local recurrence), tumors diffusely infiltrating and recurring at a distant area away from the cavity (diffuse invasion and distant recurrence), and tumors showing meningeal and intraventricular dissemination (dissemination) (Figure 5a,b). We found no significant correlation between PFS times on BCNU wafer therapy and total BCNU concentrations (AUC_all_) in the nine patients with relapse (Appendix A). Moreover, we found no significant difference in AUC_all_ of BCNU concentrations between local recurrence (L) (2666 ± 1257) and non-local recurrence (diffuse/distant recurrence or dissemination) (D) (1608 ± 623, *p* = 0185) (Figure 5c).

### 3.8. Time Courses of Protein Concentrations and the Number of Cells in the CSF within the Tumor Cavity in Patients with or without BCNU Wafers

Protein concentrations in the CSF within the cavity showed a rapid increase until 3 days postoperatively, and then gradually continued to increase until 30 days postoperatively in most patients. Two patients (G2, G4) did not show an increase in the protein concentration 14 days after the operation (Figure 6a). The LN group was thought to have lower concentrations of protein than the ER group, but the difference between the two groups was not significant (Figure 6c,d). On the other hand, concentrations of protein in the CSF within the cavity of patients without wafer treatment (control) (Figure 6b) were 1/3 to 1/5 of that in wafer-implanted cases.

The number of cells in the CSF within the cavity reached a maximum 3 days postoperatively, gradually decreased thereafter, and remained higher than normal 30 days postoperatively (Figure 7). No apparent difference was observed in the time course patterns of cells in the CSF in the cavity between the ER and LN groups. The increased cells were almost all monocytes, and no tumor cells were seen.

## 4. Discussion

BCNU wafers are biodegradable copolymers impregnated with the alkylating agent carmustine [6]. Wafers containing this chemical agent are implanted in the brain after resection of tumors to kill residual tumor cells including infiltrating tumor cells by exposing these tumors to a much higher dose of BCNU. In the present study, to elucidate whether a high concentration of BCNU released from the wafers can improve PFS and OS of the patients with GBM, we measured BCNU concentrations in the tumor resection cavity for 30 days after the operation and analyzed the relationship between total concentrations of BCNU and patient survival.

After implantation of BCNU wafers, the polymeric substrate of the wafer gradually decomposes, and the internal BCNU is released. In vitro, the polymeric substrate in the wafer decomposes into two phases [15]. During the initial 10 h (induction phase), the polymer is hydrolyzed, and during the subsequent erosion phase, the soluble polymer fraction is released and dissolves from the outer layer to the inner layer. Subsequently, BCNU is released due to wafer degradation [15,16]. As a result, 60% of BCNU is released by diffusion during the initial 10-h period (induction phase), and the remaining 40% is released over the next 5 days. Regarding the distribution of released BCNU in the human brain, few data have been published about the release and distribution of BCNU in clinical cases [17]. The present study showed that the concentration of BCNU in the tumor resection cavity was the highest (2.5–3.0 μmol/L) at day 0, rapidly decreased to 0.95–1.5 μmol/L within 24 h, was maintained at more than 0.6–0.8 μmol/L for 7 days, and decreased to an almost undetectable level (under 0.05 μmol/L) after 14 days postoperatively. This time course of BCNU concentrations in the cavity coincided well with the release patterns of BCNU from the wafer into the blood stream in previous studies [18]. These authors reported that the peak serum concentration of BCNU was 0.55 μmol/L and that the concentration decreased to 0.013 μmol/L 24 h after implantation of the wafer. Consequently, the concentration of BCNU in the CSF may be about 70-fold greater than that in serum until 7 days after BCNU wafers are implanted. The wafer itself remains on images for several months or more, but most internal BCNU is thought to be released within 1 week after implantation [19].

AUC_all_ was calculated from the time course curve of BCNU concentrations from day 0 to day 30 after the operation in each patient with GBM. Then, the correlation between AUC_all_ and PFS/OS in 11 GBM patients (1 patient was excluded due to ventricular opening) was examined, but no correlation was found. Therefore, we looked for a difference in the mean AUC_all_ of BCNU concentrations between the early progression (ER) group and late or no recurrence (LN) group. The present results indicate that the high concentration of BCNU, which was achieved by treatment with BCNU wafers, did not have a sufficient cytocidal effect to inhibit the early progression of tumors in about half of patients with GBM.

Previously, we reported that the highly invasive type of GBM on MRI has much worse prognosis than the low-invasive and demarcated type of GBM on MRI [7]. In the present study, all patients in the ER group showed the highly invasive type, and all patients in the LN group showed the low-invasive and demarcated type. These phenotypic features of GBM on MRI completely coincided with the degree of CD44 expression in the tumor tissues (P/C ratio). CD44 is a multi-functional cell surface adhesion receptor involved in regulating the progression, invasion, and resistance to radiochemotherapy of cancer cells and is regarded as a cancer stem cell marker [20,21]. In GBM, it has been reported that high expression of CD44 shows worse prognosis in the most of patients with GBM [22,23]. We reported that the values of the ratio of CD44 expression in the tumor periphery to CD44 expression in the core (P/C ratio) correlated well with the GBM phenotypes on MRI and the prognosis of patients with GBM [7]. The present results indicate that the highly invasive type of GBM on MRI may show resistance to treatment with BCNU wafers. Resistance to BCNU chemotherapy is also affected by the presence of MGMT. However, the present study found no significant differences in the degree of MGMT promotor methylation between ER and LN groups. Consequently, expression of MGMT was not related to resistance to BCNU chemotherapy in the present study. We found that glioma stem-like cells with a high expression of CD44 exist in the tumor invasion zone of the tumor periphery in highly invasive GBM [7]. As cancer stem cells are known to express a high level of CD44 [24], glioma stem cells highly expressing CD44 may participate in the resistance to even the high-dose chemotherapy with BCNU.

In the present study, the LN group (all low-invasive, demarcated type) showed significantly longer PFS and OS compared with ER group (all highly invasive type). To investigate whether differences in survivals between low-invasive and highly invasive types of GBM, are attributable to BCNU wafers, we analyzed the effect of BCNU wafers on survival in 38 patients with GBM who were treated in the same institute and under the same protocol as the present study. Of these, 18 were treated with BCNU wafers (Group 1) and 20 were not (Group 2). We also used MRI/PET to classify patients into two phenotypes: a highly invasive type and a low-invasive, demarcated type. The former type showed significantly higher expression of CD44 than the latter type in the tumor periphery of GBM. The study demonstrated that there were no significant differences in PFS and OS between Group 1 and Group 2 (Appendix A). In contrast, in Group 1, low-invasive type GBM showed significantly longer PFS and OS than highly invasive type GBM (Appendix A), In addition, in low-invasive type GBM, the treatment with BCNU wafers provided significantly longer survival times than the treatment without BCNU wafers (Appendix A). These data support the present results that BCNU wafers may provide a beneficial effect in prolonging tumor PFS and OS in low-invasive type GBM expressing CD44 at a low level. In addition, even high-dose BCNU would not be able to control the tumor cells of highly invasive type GBM expressing high CD44. These results also indicate that evaluation of the role of BCNU wafers may require consideration of different types of tumor cells showing different responses to BCNU. Investigation of the level of CD44 expression in the tumor tissues may be useful to predict the effectiveness of BCNU wafers in prolonging survival for patients with GBM.

We also examined whether high doses of BCNU affect the recurrence pattern of GBM. Before TMZ was introduced, more than 95% of GBMs were thought to recur at the periphery of the tumor resection cavity or within 20 to 30 mm from the limb of the resection cavity [25]. After TMZ was introduced, Oh et al. reported that 58 of 67 patients (87%) treated with the Stupp regimen showed local recurrence [26]. Giese et al. compared the imaging pattern of recurrence after BCNU implantation in 11 patients with that after placebo wafer implantation in 13 patients [27]. The recurrence patterns after treatment with BCNU wafers or the placebo were classified into four types. In the BCNU wafer group, eight patients developed local recurrence (72.7%), and three developed diffuse/multifocal recurrence (27.3%). In the placebo wafer group, nine developed local recurrence (69.2%), and four were diffuse (30.8%). No significant difference was found in the recurrence patterns between treatment with BCNU wafers and placebo wafers. These data indicate that BCNU wafers may not affect both tumor development and tumor progression at recurrence. In our study, nine patients recurred in one of three patterns, including local recurrence in six patients (67%), diffuse/distant recurrence in two (22%), and dissemination in one (11%). We found no significant difference in AUC_all_ of the BCNU concentration between the six patients with local recurrence and the remaining three patients with non-local recurrence.

In addition to BCNU, we measured protein concentrations and the number of cells in the tumor resection cavity at the same time course of BCNU measurement. Compared with the control (no wafers), the protein concentrations rapidly increased within 3 days after implantation of the wafers and continued to increase gradually up to 30 days postoperatively, with various degrees of protein concentrations among patients. Whether these time courses of protein concentration are related to cytocidal effects of BCNU is unclear, but four patients (67%) in the ER group continued to show high concentrations of protein up to 30 days after implantation. On the other hand, four patients (67%) in the LN group showed much lower concentrations of protein. In these patients, the mean protein concentration at 14 days postoperatively (216.5 mg/dL) was not different from the mean protein concentration in controls at 14 days postoperatively (169.5 mg/dL). In contrast, the mean protein concentration at 14 days postoperatively in three patients in the ER group was 2537.7 mg/dL. These data suggest that the protein concentration in the cavity may be affected by the number of residual tumor cells that show resistance to a high dose of BCNU.

In contrast, the time course of the number of cells in the cavity showed different curves from the protein concentration curves. The number of cells in the cavity increased until 3–7 days after implantation, but afterwards, the cell number decreased to almost normal levels at 30 days after implantation. Both the ER and LN groups showed the same time courses, suggesting that the increased protein concentrations in the ER group are not due to an inflammatory reaction to BCNU wafers.

The present study consisted of a small number of patients. To obtain a more definite conclusion, more extensive analysis with an increased number of patients will be required. As mentioned above, our recent analysis in 38 patients with GBM who were treated with the same protocol supported the results of the present study. Although BCNU wafers may not provide a beneficial effect on the prognosis in highly invasive phenotype of GBM, the treatment may have potential to improve the survival in a low-invasive phenotype of GBM expressing CD44 at a low level in the tumor periphery of GBM.

## 5. Conclusions

To investigate whether local high-dose chemotherapy by BCNU wafers is effective in the prognostic outcome in malignant glioma, particularly GBM, we measured BCNU concentrations in the tumor resection cavity. BCNU concentrations in the cavity showed a peak value 1 h after implantation of BCNU wafers, rapidly decreased within 24 h, and were maintained at a relatively high concentration for 7 days after implantation. AUC_all_ representing total BCNU concentrations in the cavity for 30 days after implantation of BCNU wafers was higher in patients showing late or no recurrence than in those showing early tumor recurrence, but the difference was not significant. Patients showing late or no recurrence had much longer survival than those showing early recurrence; in particular, the former patients showed remarkably longer median PFS time. Imaging analysis showed that phenotypic features of GBM on MRI completely coincided with the time of tumor recurrence. That is, the highly invasive type of GBM on MRI showed early recurrence after treatment with BCNU wafers, whereas the low-invasive and demarcated type of GBM showed late or no recurrence. These results suggest that local treatment with BCNU wafers may have potential to improve the prognosis of patients whose tumor is the low-invasive and demarcated type of GBM. However, even locally enhanced BCNU concentrations may not be effective to inhibit the growth of all of GBMs. Selection of patients who are likely to respond well to the chemotherapeutic agent is important to obtain optimum treatment for patients with GBM.

## Figures and Tables

**Figure 1 brainsci-12-00567-f001:**
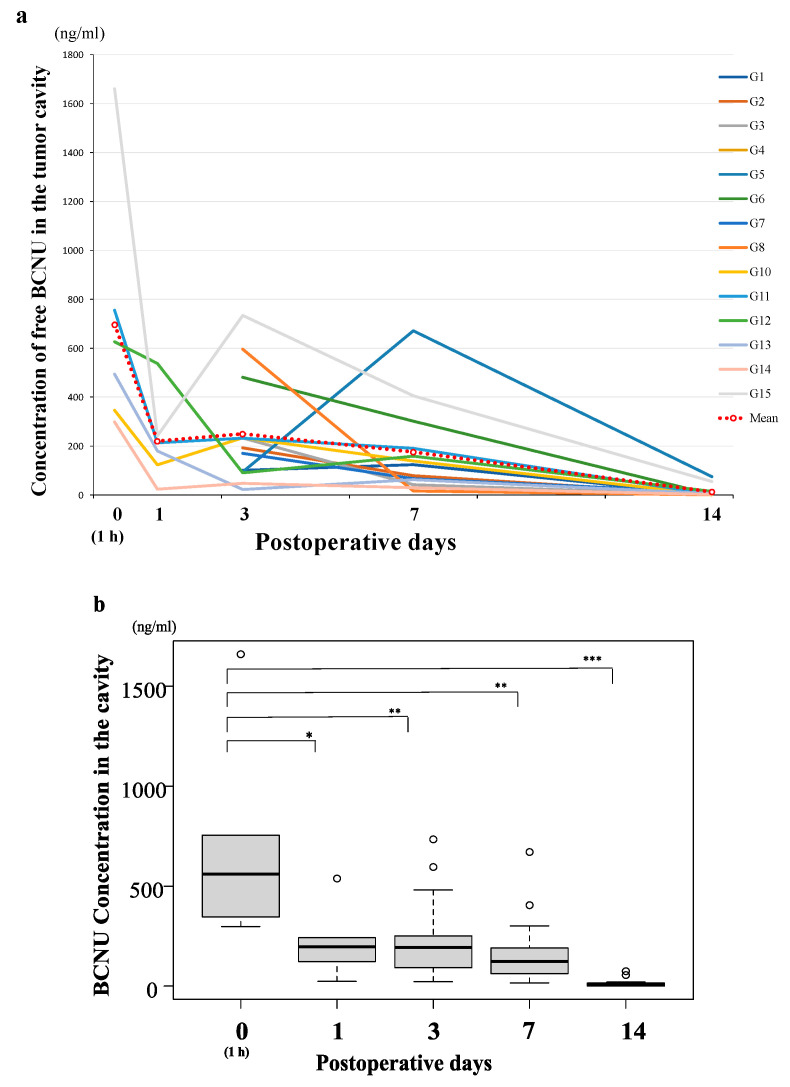
Time course of the bis-chloroethyl nitrosourea (BCNU) concentration (days 0–14) in the tumor resection cavity in 14 patients with malignant glioma treated with BCNU wafers. (**a**) The concentration curves of BCNU showing the sampling data of 14 patients. The mean BCNU concentration at each sampling time was calculated and presented in the graph (red dotted line). (**b**) Time course of the median BCNU concentration shown in box plots represented as horizontal bars, as well as the 5th and 95th percentiles. The BCNU concentration was the highest (mean/median (Mn/Md): 697/560 ng/mL) at day 0 (1 h), rapidly decreased to 220/196 (Mn/Md) ng/mL within 24 h, was maintained at more than 176/124 (Mn/Md) ng/mL for 7 days, and decreased to an almost undetectable level at 14 days after implantation. The concentration of BCNU at day 0 was significantly higher than that at day 3, day 7, and day 14 (* *p* < 0.01, ** *p* < 0.001, *** *p* < 0.0001).

**Figure 2 brainsci-12-00567-f002:**
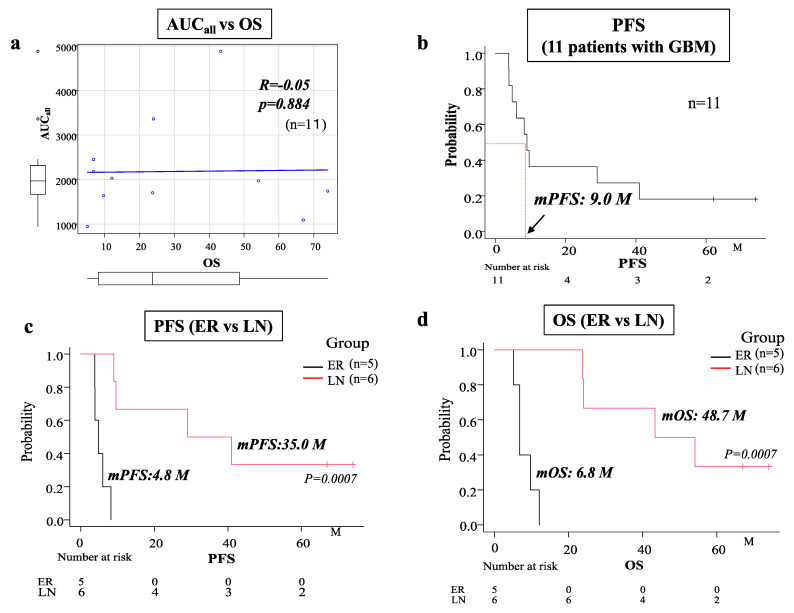
(**a**) Spearman’s regression analysis showing no correlation between total concentration (AUC_all_) of BCNU and overall survival (OS) in 11 patients with GBM. (**b**) Kaplan–Meier survival curve showing progression-free survival (PFS) in 11 patients with GBM (the median PFS was 9 months). (**c**,**d**) Kaplan–Meier survival curves showing PFS (**c**) and OS (**d**) in the early recurrence (ER) group (five patients) and the late or no recurrence (LN) group (six patients). Patients with a median PFS < 9 months were classified in the ER group, whereas patients with a median PFS ≥ 9 months were included in the LN group. The median PFS of the ER group (4.8 months) was significantly shorter than that of the LN group (35.0 months) (*p* = 0.0007). The median OS of the LN group (48.7 months) was significantly longer than that of the ER group (6.8 months) (*p* = 0.0007).

**Figure 3 brainsci-12-00567-f003:**
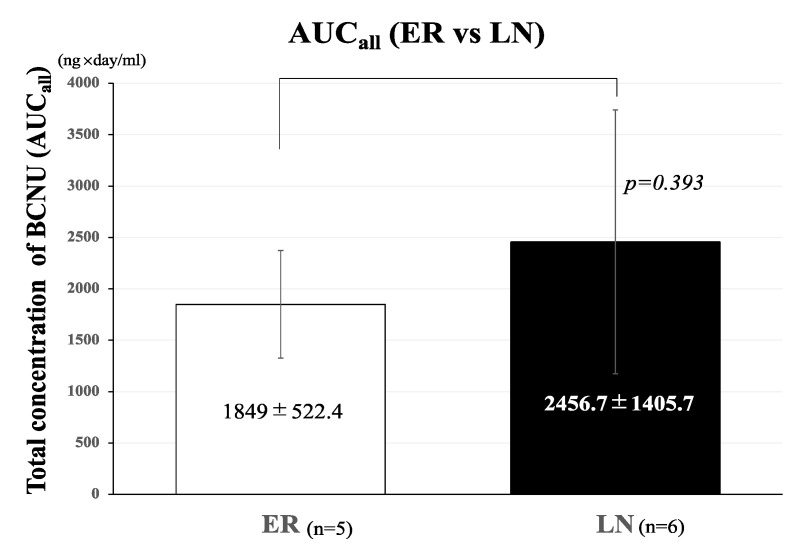
Effects of total concentration (AUC_all_) of BCNU on the development of tumor recurrence. Bar graph showing the comparison of AUC_all_ of BCNU between the ER and LN groups. The AUC_all_ of BCNU concentrations in the ER group (1849 ± 552.4 ng·day/mL) was lower than that in the LN group (2456.7 ± 1405.7 ng·day/mL), but the difference was not statistically significant (*p* = 0.393). The bar graph shown in a white color represent AUC_all_ of ER and that in a black color represent AUC_all_ of LN. Error bars represent standard deviation.

**Figure 4 brainsci-12-00567-f004:**
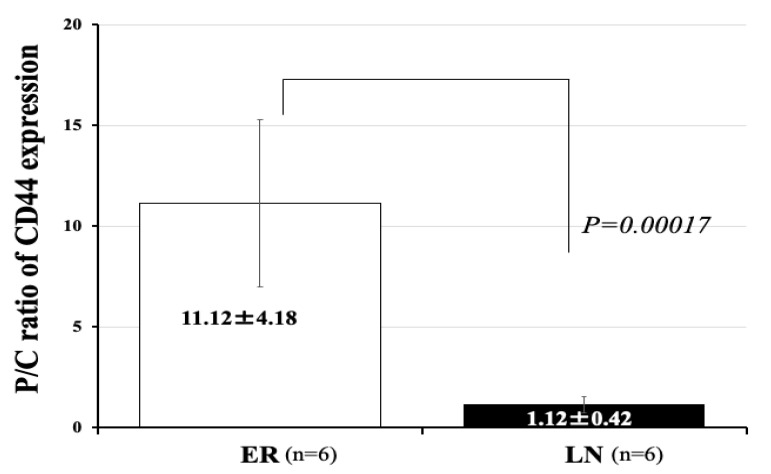
Expression of mRNA of CD44 in 12 patients with GBM. Periphery/core (P/C) ratios for CD44 expression in the ER group (high invasive) and LN group (low invasive). The ER group (high invasive) showed a significantly higher P/C ratio for CD44 than the LN group (low invasive) (ER vs. LN: 11.12 ± 4.18 vs. 1.12 ± 0.42) (*p* = 0.00017). The bar graph shown in a white color represent CD44 expression of ER and that in a black color represent CD44 expression of LN. Error bars represent standard deviation.

**Figure 5 brainsci-12-00567-f005:**
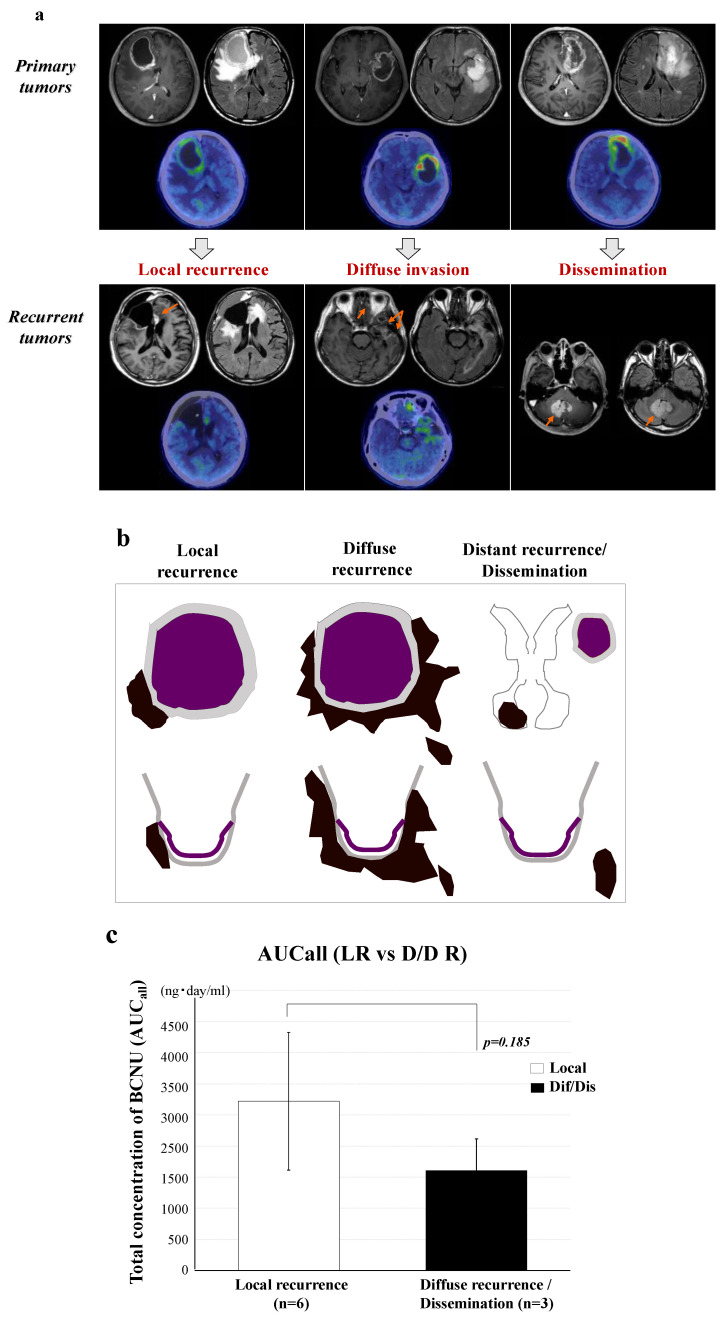
Patterns of tumor recurrence following treatment with BCNU wafers and the effects of AUC_all_ on the tumor recurrence patterns. (**a**) Tumor growth patterns at the time of tumor recurrence were classified into three types on MRI according to continuity with the areas around the resection cavity (arrows show the sites of tumor recurrence). (**b**) Illustration showing the recurrence patterns of tumors. These included tumors recurring from the edge of the tumor resection cavity (local recurrence), diffuse infiltration to a distant area (diffuse/distant invasion), and meningeal and intraventricular dissemination (dissemination). The two latter patterns were non-local recurrence. (**c**) Effects of the total concentration (AUC_all_) of BCNU on the recurrence pattern in GBM. No significant difference was present in AUC_all_ between local recurrence and non-local recurrence (*p* = 0.185). A white color bar graph shows AUC_all_ of local recurrence and a black color bar graph shows that of non-local recurrence. Error bars represent standard deviation.

**Figure 6 brainsci-12-00567-f006:**
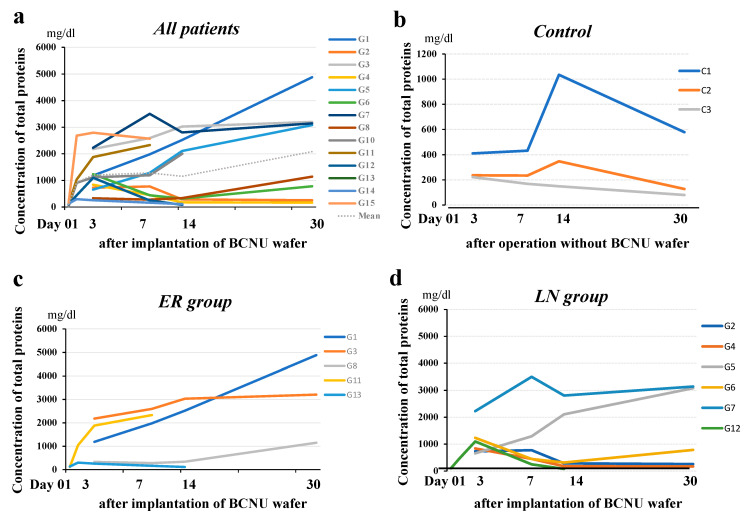
Time course of protein concentration in the resection cavity. (**a**) Protein concentrations in the resection cavity in 15 patients with malignant glioma showed a rapid increase up to 3 days postoperatively, and then a gradual increase up to 30 days postoperatively in most patients. (**b**) Protein concentrations in the resection cavity in the controls (patients with no BCNU wafer treatment). The concentrations decreased to under 350 mg/dL at 30 days after the operation. (**c**) Protein concentrations in the resection cavity in the patients in the ER group showed a tendency to increase with time. (**d**) Protein concentrations in the resection cavity in the patients in the LN group showed a tendency to decrease with time.

**Figure 7 brainsci-12-00567-f007:**
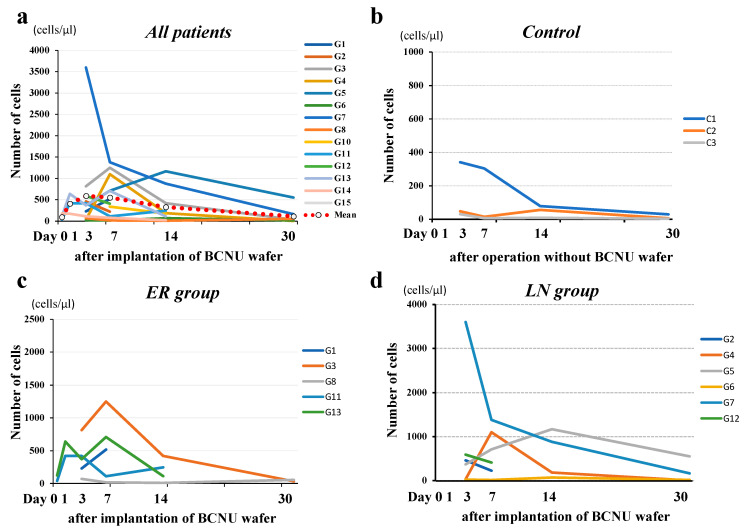
Time course of the number of cells in the resection cavity. (**a**) The number of cells in the resection cavity reached a maximum on the third postoperative day and gradually decreased thereafter, and remained higher than normal (**b**) on the 30th postoperative day. No difference in the time-course patterns of cells in the CSF in the cavity between the ER (**c**) and LN groups (**d**).

**Table 1 brainsci-12-00567-t001:** Characteristic features, outcome, concentrations of BCNU, tumor phenotypes, and expression of CD44 in the glioma patients with Gliadel wafer therapy.

Patient		KPS		Methylation of		Recurrence		Outcome		BCNU (AUCall) (ng ×	Phenotype on	CD44 Expression (mRNA)		
No.	Age/Sex	(%)	Histology	MGMT	EOR	Rec PFS (M)		D/A	OS (M)	day/mL)	MRI/PET	Core	Periphery P/C Ratio	
G1	67/M	70	GBM	+	GTR	+	8.2	D	12.1	2030	HI	1	16.3	15.9
G2	64/F	90	GBM	−	GTR	−	74	A	74	1740	LI	2.7	5.3	1.9
G3	71/M	90	GBM	−	GTR	+	6	D	9.7	1640	HI	1.1	19.2	16.8
G4	63/F	80	GBM	−	GTR	+	41	D	54.1	1970	LI	55.2	69.1	1.3
G5	79/F	70	GBM	−	PR	+	29	D	43.3	4880	LI	2.8	2.7	1
G6	30/M	70	GBM	−	GTR	+	9.6	D	24	3360	LI	2.8	2.1	0.8
G7	65/F	70	GBM	+	GTR	−	67	A	67	1090	LI	2.9	2.8	1
G8	67/M	60	GBM	+	STR	+	3.8	D	6.8	2180	HI	1.6	13	8.2
G9	86/M	80	GBM	+	GTR	+	2.4	D	2.4	2970	HI	4.1	30.8	7.5
G10	45/M	90	AMG3	−	STR	−	62.1	A	62.1	1820	na	1	1	1
G11	71/F	80	GBM	−	GTR	+	4.8	D	6.8	2450	HI	3.1	25	8
G12	65/M	70	GBM	+	STR	+	9	D	23.8	1700	LI	3.3	3.1	0.9
G13	66/M	70	GBM	−	GTR	+	3.9	D	5.1	945	HI	4	41.2	10.4
G14	30/M	70	AMG3	−	PR	−	58	A	58	491	na	0.8	0.7	0.9
G15	35/F	90	AMG3	+	GTR	−	57	A	57	5810	na	0.9	0.8	0.9
**Patients without Gliadel (control)**														
C1	59/M	70	GBM	+	PR	−	2.6	D	8.3	na				
C2	72/M	90	AMG3	−	GTR	−	30.4	A	30.4	na				
C3	64/F	80	GBM	+	GTR	+	7.4	D	23.5	na				

No., number; F, female; M, male; G, glioma; GBM, glioblastoma, IDH-wild; AMG3, astrocytoma, IDH-mutant, grade 3; MGMT, O (6)-methylguanine-. DNA methyltransferase; EOR, extent of resection; GTR, gross total resection; STR, subtotal resection; PR, partial resection; KPS, Karnofsky performance status; P/C ratio, periphery/core ratio; PFS, progression-free survival time; OS, overall survival time; M, month; +, survival time in the patients alive at the last follow-up; A, alive; D, dead; na, non-assessment; HI, highly invasive type; LI, low invasive type; P/C ratio, CD44 in the periphery/in the core.

## Data Availability

Not applicable.

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
