# Peer review of "Is Interstitial Chemotherapy with Carmustine (BCNU) Wafers Effective against Local Recurrence of Glioblastoma? A Pharmacokinetic Study by Measurement of BCNU in the Tumor Resection Cavity"

_brainsci, 2022, doi:10.3390/brainsci12050567_

Round 1

Reviewer 1 Report

From my point of view all criticism are considered sufficiently.

Reviewer 2 Report

I read the revised version of this manuscript. The paper has been improved taking into account my suggestions.

This manuscript is a resubmission of an earlier submission. The following is a list of the peer review reports and author responses from that submission.

Round 1

Reviewer 1 Report

I read with interest this article investigating the role of BCNU wafers in local recurrence of GBM. Despite different efforts the prognosis of GBM remains poor. The role of BCNU wafers in prolonging survival in GBM has been previously studied and the results are controversial. Overall there is no a clear advantage of this therapy and consequently it is has not been included in the standard protocols of care of GBM. The main shortcomings of this paper are the following:

1. only 10 GBM have been included in the protocol

2. there is no control group

3. the authors reported that in some cases there was a ventricle opening during surgery. In my opinion these patients should be excluded from the analysis

4. there was no difference in BCNU concentrations between LN group and ER group and in the ER group some patients had a GBM recurrence. Thus the conclusions of Authors: "These results suggest that local treatment with BCNU wafers may improve the prognosis in patients whose tumor is the low-invasive and demarcated type of GBM" are not justified by the presented data 

Reviewer 2 Report

The researches described in their study the effectiveness and a resistant GBM phenotype to BCNU wafers by measuring the BCNU concentration in the resection cavity of GBM and analysing the relationship between BCNU concentration and survival. BCNU wafers were implanted with an Ommaya device and BCNU concentrations in the tumour resection cavity were measured for 30 days postoperatively. 12 GBM patients were classified into two groups, early recurrence (ER) and late or no recurrence (LN) by taking median PFS as a cut- off value. They found that the total BCNU concentrations were not correlated with tumor progression or survival.

Even though the novelty and originality is not very high, the topic and the findings are very interesting for many scientists working in the research field of Neurooncology.

However I have some remarkes:

In the abstract part the authors described that they have investigated 12 GBMs. In the material and method section they describe, that they have enrolled eighteen patients (14 GBMs, 3 AA and 1 AOA). In table 1 they describe 15 patients and 3 patients as a control group.

This is very confusing and this should be clarified.

Furthermore, the histological diagnosis of an AOA doesn´t exist anymore since the revised WHO classification of 2016. As the current WHO classification of the brain tumors has been changed again in the year 2021, I would suggest that the authors match their histological findings to the latest WHO classification.

It is very surprising that all tumors were negative concerning the MGMT status of the given tumors, because nearly half of the GBMs in the literature are MGMT methylated. This suggests that there is a problem in their investigation of the MGMT status.

Without a clear histological diagnosis and a clear MGMT status of the given tumors on comments could be made concerning the clinical outcome of the patients.